

# Age-related changes in the distributions of depressive symptom items in the general population: a cross-sectional study using the exponential distribution model

Shinichiro Tomitaka[1,2], Yohei Kawasaki[2], Kazuki Ide[2], Hiroshi Yamada[2], Toshiaki A. Furukawa[3] and Yutaka Ono[4]

[1] Department of Mental Health, Panasonic Health Center, Tokyo, Japan

[2] Department of Drug Evaluation and Informatics, Graduate School of Pharmaceutical Sciences, University of Shizuoka, Shizuoka, Japan

[3] Department of Health Promotion and Human Behavior, Department of Clinical Epidemiology, Kyoto UniversitGraduate School of Medicine/School of Public Health, Kyoto University, Kyoto, Japan

[4] Center for the Development of Cognitive Behavior Therapy Training, Tokyo, Japan

Corresponding author
Shinichiro Tomitaka, tomitaka.shinichiro@jp.panasonic.com

## ABSTRACT

**Background.** Previous research has reported inconsistent evidence of the trajectory of depressive symptoms across the adult lifespan. We investigated how the distributions of each item score change with age and determined whether the trajectory of depressive symptoms varied with the scoring methods of the questionnaire.

**Methods.** We analyzed data collected from 21,040 subjects who participated in the national survey in Japan. Depressive symptoms were assessed using the Center for Epidemiologic Studies Depression Scale (CES-D). The CES-D has 20 items, each of which is scored in four grades of "rarely," "some," "much," and "most of the time." We used the exponential distribution model which fits the distributions of 16 negative symptom items of CES-D, with the probabilities of "some," "much," "most," and "rarely" expressed as $P$, $Pr$, $Pr^2$, and $1 - P \times (r^2 + r + 1)$.

**Results.** The distributions of the responses to 16 negative symptom items followed the common exponential model across all age groups. The mean of the estimated parameter $r$ of 16 negative items showed a U-shape pattern, being high during 12–29 years, remaining low during 30–50 years, and then increasing again over 60 years. The trajectory of depressive symptom scores simulating the binary method was different from that of the empirical scores using the Likert method.

**Conclusions.** Our findings show that the increase in the depressive symptoms score during older age is based on the increase of the parameter $r$. The differences in the scoring method may contribute to the different age-related patterns across the adult lifespan.

## INTRODUCTION

Depression, a common mental disorder, is one of the leading causes of disability world-wide (*Moussavi et al., 2007*). There has been much interest in understanding the distribution of depressive symptoms in the general population, given their association with depression (*Blazer & Kessler, 1994*; *Kroenke et al., 2009*).

Cross-sectional surveys and longitudinal studies, the majority of which made using the Center for Epidemiologic Studies Depression Scale (CES-D), have found that the trajectory of depressive symptoms across the adult lifespan follows a U-shaped pattern, with symptoms being high during young adulthood, decreasing during middle adulthood, and then increasing again over the age of 70 (*Kessler et al., 1992*; *Oh et al., 2013*; *Sutin et al., 2013*; *Tomitaka, Kawasaki & Furukawa, 2015a*). Conversely, some studies, particularly cross-sectional surveys using the Revised Clinical Interview Schedule (CIS-R), consistently found that depressive symptoms were high during middle adulthood and decreased over the age of 55 (*Adult psychiatric morbidity in England, 2007*; *Bromley et al., 2011*).

The reason for the differences observed in age-related manifestations of depressive symptom between CES-D and CIS-R is unclear. It is unlikely that the true effect of aging on depressive symptoms in the general population is inconsistent, indicating a possible influence of methodological factors on the results of epidemiological studies.

The discrepancy between epidemiological studies could be explained by selection bias. However, this explanation conflicts with the fact that even the large sample surveys in the randomly selected general population consistently show differences between CES-D and CIS-R (*Adult psychiatric morbidity in England, 2007*; *Tomitaka, Kawasaki & Furukawa, 2015a*).

Another possible explanation is the difference in choice of items between CES-D and CIS-R. The CES-D consists of 16 negative symptom items and 4 positive affect symptom items (good, hopeful, happy, and enjoyed) that are not included in CIS-R (*Radloff, 1977*). Conversely, CIS-R contains 14 negative symptom items (somatic symptom, fatigue, concentration, sleep, irritability, worry about physical health, depression, depressive ideas, worry, anxiety, phobias, panic, compulsions, and obsessions), some of which (e.g., panic, compulsion, and obsession) are not included in CES-D (*Lewis et al., 1992*) However, even the common items (e.g., depression, depressive ideas, sleep, anxiety, and fatigue) show differences in age-related patterns between CES-D and CIS-R (*Adult psychiatric morbidity in England, 2007*; *Bromley et al., 2011*; *Cooper et al., 2015*). Therefore, it is difficult to attribute these differences between CES-D and CIS-R to the choice of items; it is necessary to consider why even the same items often differ.

CES-D and CIS-R differ in the scoring method used for each item. In CES-D, each item has one question which is scored using a four point Likert method (0-1-2-3) based on four possible answers: "rarely or none of the time (rarely)," "some or little of the time (some)," "occasionally or a moderate amount of time (much)," and "most or all of the time (most)" (*Radloff, 1977*). Conversely, each item of CIS-R has two mandatory questions to screen the specific symptom (*Lewis et al., 1992*). If responses indicate the specific symptom, four questions are asked and scored using a binary method (0–1). The
four non-mandatory questions contribute a single point to the 0–4 scale for each item (except depressive idea which has a 0–5 scale). The severity of each symptom appears to weigh more heavily in the CES-D scale than in the CIS-R scale. These differences in the scoring method may contribute to the different age-related patterns observed across the adult lifespan. In support of this speculation, other studies investigating binary method questionnaires (e.g., Bradburn Affect Balance Scale) also reported that negative affect symptom items were higher during middle adulthood than during old age (*Stacey & Gatz, 1991*; *Charles, Reynolds & Gatz, 2001*). To elucidate the effects of different scoring methods on each item score, it is necessary to understand how the distributions of each item score change with age. Therefore, a method for representing the distribution is necessary in order to detect age-related changes.

Using a variety of multivariate statistical methods, e.g., factor analysis, principal components analysis and cluster analysis, a lot of population studies on depressive symptoms have been performed so far. A common feature of these multivariate statistical methods is that they are focused on the interrelationship of each variable. Even though the interrelationship of each variable has been overwhelmingly investigated by many researchers, to the best of our knowledge, little work has been done to elucidate the mathematical patterns of the distribution of each depressive symptom. Thus, in the previous study, we investigated the mathematical patterns of the individual distributions for each item of the CES-D.

Although the method of the previous study was simple (just a histogram), we found an intriguing phenomenon. The distributions of the 16 negative items for CES-D commonly exhibited exponential patterns between "some" and "most" response levels, while "rarely" was not related to the exponential patterns between "some" and "most" (*Tomitaka, Kawasaki & Furukawa, 2015b*). Based on the findings, we proposed a mathematical model for the distribution of the 16 negative symptom items, which were expressed by two parameters, $P$ and $r$, where $P$ stands for the probability of "some" and $r$ stands for the equal ratio among "some," "much," and "most." The probabilities of "some," "much," "most," and "rarely" are expressed as $P$, $Pr$, $Pr^2$, and $1 - P \times (r^2 + r + 1)$. We utilized this mathematical model to quantify the age-related changes in the distribution of negative symptom items.

The present study used cross-sectional data from a large, nationally representative survey conducted annually to evaluate the health status of a representative sample from the general Japanese population (*Ministry of Health, Labor and Welfare, Statistics and Information Department, 2002*). We analyzed more than 20,000 CES-D assessments performed in subjects between 12 and 89 years.

Here we investigated whether the age-related changes observed in the total CES-D score were mainly due to the 16 negative symptoms. After confirming that the 16 negative symptoms were responsible for the U-shaped pattern of the total CES-D score and the distribution of the 16 negative items followed the mathematical model across all age groups, we used the mathematical model to examine how the distribution of these symptoms changed according to the age group. Finally, we tested whether the conversion

of each item score of CES-D from the standard Likert method to the binary method could affect age-related changes in depressive symptoms across adulthood.

## METHODS

This study used data from the Active Survey of Health and Welfare (ASHW) conducted by the Japanese Ministry of Health, Labor and Welfare in 2000 (*Ministry of Health, Labor and Welfare, Statistics and Information Department, 2002*). ASHW is an annual nationwide survey conducted by the Japanese Government to collect data necessary for policy making and health promotion in compliance with the Statistics Law. The legal and ethical approval of ASWH was given by the Ministry of Health, Labor and Welfare, Japan. In 2000, ASHW examined depressive symptoms among a representative sample from the general Japanese population. To ensure that the sample was adequately representative, survey participants were selected from individuals aged >12 years living in 300 communities in Japan. These communities were selected from 881,851 precincts identified in the 1995 Census using a stratified sampling design. Verbal informed consent was obtained from all the subjects. The data and methods used by the survey have been described in detail before (*Ministry of Health, Labor and Welfare, Statistics and Information Department, 2002*).

The questionnaire was returned by 32,729 respondents, and the response rate was not publicized by the Ministry of Health, Labor and Welfare and Health. However, the response rates for similar surveys conducted 3 and 4 years earlier were 87.1% and 89.6%, respectively (*Kaji et al., 2010*). Therefore, we assumed that the response rate for the present survey was over 80%.

The present study was secondary analysis of ASWH data. The Ministry of Health, Labor and Welfare examined our study and permitted us to analyze data from ASWH in compliance with the Statistics Law. The present study was approved by the ethics committee of Panasonic Health Center (approval number, 2014-1), and the procedures were carried out in accordance with the approved guidelines and regulations.

We excluded 1,394 respondents as we suspected the validity of their responses (i.e., those who answered "rarely" or "most" for all items, regardless of the nature of the item). A total of 9,588 respondents with missing information on one or more key study variables (i.e., depressive symptoms or age or sex) were also excluded from the sample. An additional 50 respondents over 89 years of age were excluded as the number was too small to elucidate the pattern of distribution. The final sample consisted of 20,990 respondents between 12 and 89 years.

## MEASURES

Depressive symptoms were assessed using the Japanese version of the CES-D (*Shima et al., 1985*). This 20-item scale assesses the frequency of a variety of depressive symptoms within the previous week (0 = rarely or none of the time (less than 1 day), 1 = some or little of the time (1–2 days), 2 = occasionally or a moderate amount of time (3–4 days), and 3 = most or all of the time (5–7 days)), yielding a total score of 0–60 (*Radloff, 1977*). Higher scores indicate greater psychological distress. The 20 items of the CES-D were grouped into

the following four subscales: depressive mood (items 3, 6, 9, 10, 14, 17, and 18), somatic and retarded activities symptoms (items 1, 2, 5, 7, 11, 13, and 20), interpersonal relations (items 15 and 19), and positive affect symptoms (items 4, 8, 12, and 16). The positive affect symptom items were reverse-scored. The 16 items of depressive mood, somatic and retarded activities symptoms, and interpersonal relations follow the common mathematical model, while the 4 positive affect symptom items do not (*Tomitaka, Kawasaki & Furukawa, 2015b*).

## Analysis procedure

The respondents were grouped into the following age groups: 12–19, 20–29, 30–39, 40–49, 50–59, 60–69, 70–79, and 80–89 years. The final sample consisted of 20,990 respondents between the ages of 12–89 years (ages 12–19; $N = 2,457$ (male; $n = 1,269$), ages 20–29; $N = 3,748$ (male; $n = 1,788$), ages 30–39; $N = 3,761$ (male; $n = 1,783$), ages 40–49; $N = 3,629$ (male; $n = 1,788$), ages 50–59; $N = 3,569$ (male; $n = 1,800$), ages 60–69; $N = 2,253$ (male; $n = 1,155$), ages 70–79; $N = 1,161$ (male; $n = 517$), ages 80–89; $N = 412$ (male; $n = 108$)).

The first step was to compare the total CES-D score and each item score by age group, and to confirm that the 16 negative symptom items followed a U-shaped pattern across adulthood, resulting in the U-shaped pattern of the total CES-D score. This pattern of each negative item was shown to exhibit the lowest score during 30–69 years (*Tomitaka, Kawasaki & Furukawa, 2015a*).

Second, in order to confirm that the distribution of the 16 negative items follow the mathematical model across all age groups, their histograms were evaluated for each age group. The distributions between "some" and "most" were analyzed using a log-normal scale.

Third, the parameters $P$ and $r$ were estimated from empirical data to quantify the age-related changes in the distribution of the 16 negative items. These parameters were assessed for each item using the mathematical model as follows: $P$ = probability of "some," $r$ = (probability of "much")/(probability of "some") + (probability of "most")/(probability of "much")/2. The mean values of the parameters were calculated for each age group.

Finally, to demonstrate the effect of the scoring methods on age-related changes in depressive symptoms, the total CES-D score and 16 negative items scores were compared between the standard Likert method and the binary method. We used JMP version 11 for Windows (SAS Institute, Inc., Cary, NC, USA) to calculate the descriptive statistics and frequency distribution curves.

## RESULTS

### Age-related changes of each item score and total CES-D score across adulthood

The total scores of 20 items and 16 negative item scores exhibited a similar U-shaped pattern, with symptoms being highest during 12–29 years, decreasing during 30–69 years, and then increasing again during 70–89 years, whereas the 4 positive items showed a plateau-shaped pattern (Fig. 1). The similarity in the pattern of the total score of 20 items
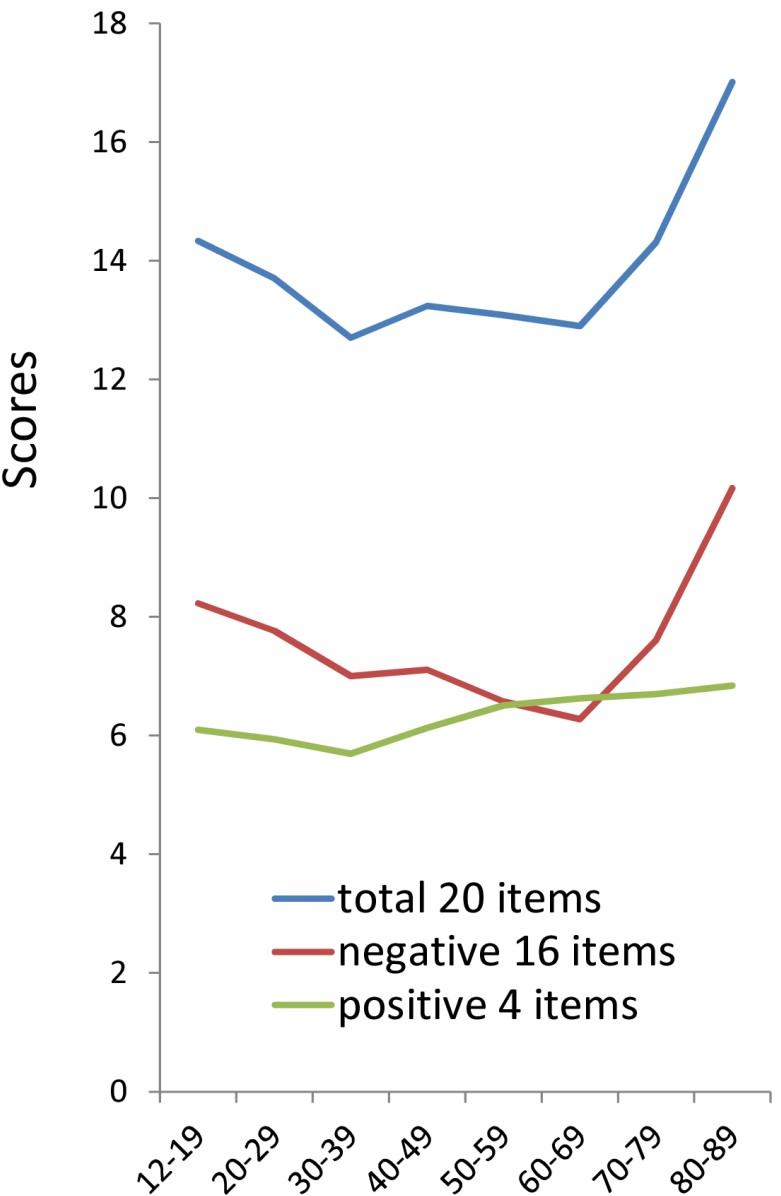

**Figure 1** **The relationship between age and the total scores of 20 items, 16 negative items score, and 4 positive items score.** The total scores of 20 items (blue line) and 16 negative item scores (red line) exhibited a similar U-shaped pattern, whereas the 4 positive items (green line) showed a plateau-shaped pattern.

and the 16 negative items score suggested that the U-shaped pattern of total CES-D score is mainly attributed to the age-related changes of the 16 negative symptom items.

Analysis of each item score of CES-D by age group showed that most of the 16 negative items in the depressive mood group, somatic symptoms and retarded activities group, and interpersonal relations group exhibited U-shaped patterns (Table 1). All 16 negative symptom items were the highest during 12–19 years or 80–89 years and 13 negative items were the lowest during 30–69 years, indicating that the U-shaped pattern is mostly common

**Table 1 Mean of each item according to age group in the general Japanese population.**

| Number | Age group | 12–19 | 20–29 | 30–39 | 40-49 | 50–59 | 60–60 | 70–79 | 80–89 |
|---|---|---|---|---|---|---|---|---|---|
| | | | | *Depressed mood* | | | | | |
| 3 | Blues | 0.41 | 0.40 | 0.39 | 0.42 | 0.39 | [a]0.33 | 0.44 | [b]0.63 |
| 6 | Depressed | 0.79 | 0.79 | 0.71 | 0.70 | 0.59 | [a]0.47 | 0.55 | [b]0.79 |
| 9 | Failure | 0.74 | 0.73 | 0.66 | 0.63 | [a]0.61 | 0.65 | 0.71 | [b]0.77 |
| 10 | Fearful | 0.27 | 0.27 | [a]0.24 | 0.28 | 0.28 | 0.28 | 0.30 | [b]0.41 |
| 14 | Lonely | 0.33 | 0.42 | 0.30 | 0.29 | 0.28 | [a]0.28 | 0.41 | [b]0.72 |
| 17 | Crying | 0.14 | 0.15 | 0.11 | 0.10 | [a]0.08 | 0.09 | 0.12 | [b]0.17 |
| 18 | Sad | 0.39 | 0.41 | 0.34 | 0.34 | [a]0.32 | 0.32 | 0.35 | [b]0.49 |
| | | | | *Somatic symptoms and retarded activities* | | | | | |
| 1 | Bothered | [a]0.54 | 0.61 | 0.67 | 0.71 | 0.65 | 0.59 | 0.69 | [b]0.89 |
| 2 | Appetite | 0.41 | 0.43 | 0.39 | 0.37 | 0.33 | [a]0.32 | 0.46 | [b]0.67 |
| 5 | Trouble concentrating | 0.95 | 0.72 | 0.66 | 0.66 | 0.58 | [a]0.52 | 0.70 | [b]0.86 |
| 7 | Effort | 1.05 | 0.86 | 0.77 | 0.75 | 0.64 | [a]0.59 | 0.75 | [b]1.11 |
| 11 | Sleep | [a]0.46 | 0.55 | 0.47 | 0.49 | 0.57 | 0.65 | 0.77 | [b]0.82 |
| 13 | Talked | [a]0.41 | 0.47 | 0.49 | 0.54 | 0.49 | 0.49 | 0.52 | [b]0.60 |
| 20 | Get going | [b]0.78 | 0.44 | 0.35 | 0.34 | [a]0.27 | 0.28 | 0.38 | 0.54 |
| | | | | *Interpersonal relations* | | | | | |
| 15 | Unfriendly | 0.26 | 0.26 | 0.24 | 0.26 | [a]0.25 | 0.24 | 0.27 | [b]0.38 |
| 19 | Dislike | [b]0.31 | 0.24 | 0.22 | 0.23 | 0.22 | [a]0.18 | 0.19 | 0.30 |
| | | | | *Positive affects* | | | | | |
| 4 | Good | 1.58 | 1.46 | [a]1.42 | 1.46 | 1.67 | [b]1.74 | 1.70 | 1.72 |
| 8 | Hopeful | 1.74 | 1.52 | [a]1.47 | 1.60 | 1.67 | 1.68 | 1.75 | [b]1.87 |
| 12 | Happy | 1.63 | 1.64 | [a]1.53 | 1.63 | 1.72 | 1.75 | [b]1.76 | 1.71 |
| 16 | Enjoyed | [a]1.15 | 1.31 | 1.28 | 1.43 | 1.45 | 1.46 | 1.50 | [b]1.54 |
| | Total 16 negative items | 8.2 | 7.8 | 7.0 | 7.1 | 6.6 | [a]6.3 | 7.6 | [b]10.2 |
| | Total 4 positive items | 6.1 | 5.9 | [a]5.7 | 6.1 | 6.5 | 6.6 | 6.7 | [b]6.8 |
| | Total 20 items | 14.3 | 13.7 | 12.7 | 13.2 | 13.1 | [a]12.9 | 14.3 | [b]17.0 |

**Notes.**
[a]The lowest value.
[b]The highest value. The positive symptom items are reverse-scored.

to the 16 negative items. Only three negative items belonging to the somatic and retarded activities symptoms subscale (i.e., "bothered," "sleep," and "talked") did not show the U-shaped pattern. "Bothered," "sleep", and "talked" were the lowest during 12–19 years and highest during 80–89 years.

In contrast to the negative items, the relationship between age and positive affect symptom items were mild and differed from each other. "Good" was the highest during 60–69 years, "happy" was the highest during 70–79 years, and "hopeful" and "enjoyed" were the highest during 80–89 years.

## Confirming that the distribution of 16 negative items follow the mathematical model

To confirm that they followed the mathematical model., the distributions of the 16 negative items were evaluated among each group (12–19, 20–29, 30–39, 40–49, 50–59, 60–69, 70–79, and 80–89 years) (Figs. 2A–2H).

The distribution of the 16 items showed a common pattern across all age groups (Fig. 2). The lines for the 16 items seemed to intersect at a single point between "rarely" and "some" across all age groups. As previously reported, all the lines that follow this

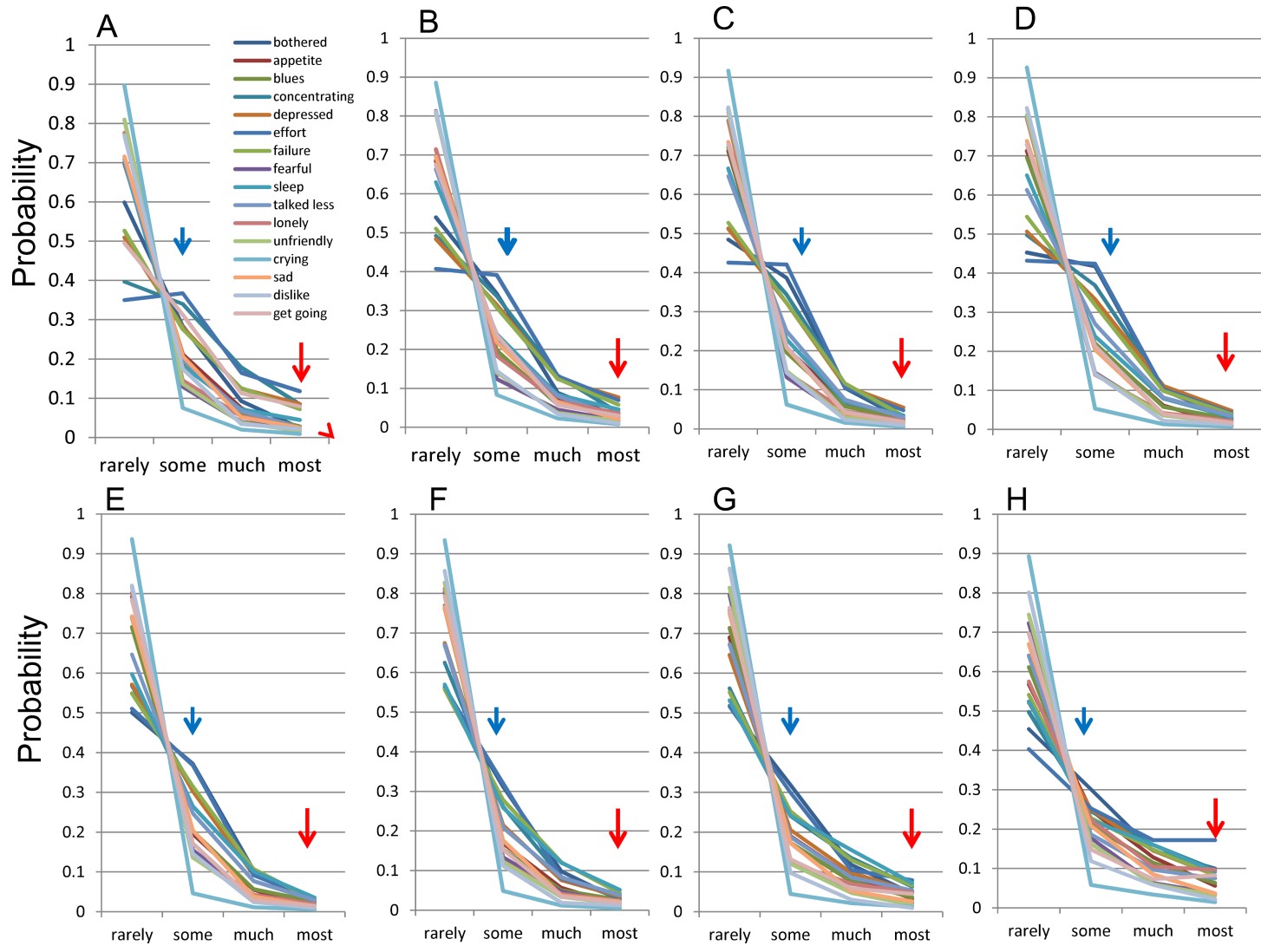

**Figure 2** **The distributions of 16 negative symptoms among each group.** (A) 12–19, (B) 20–29, (C) 30–39, (D) 40–49, (E) 50–59, (F) 60–69, (G) 70–79, and (H) 80–89 years. Red arrows indicate the probability of "most" and blue arrows indicate the probability of "some." Although the distributions for each of the 16 items followed a common mathematical pattern across all age groups, the graphs of the distributions appeared to change with age.

mathematical model theoretically cross at a single point between "rarely" and "some" (*Tomitaka, Kawasaki & Furukawa, 2015b*). Conversely, the lines for "some" to "most" responses were regularly skewed towards "some".

Although the distributions for each of the 16 items followed the common mathematical model across all age groups, the graphs of the distributions appeared to change with age. This change in the distribution with age was examined by focusing on "some" and "most" responses. As the red arrows indicate, the frequencies of "most" response for 16 items were decreasing during 12–39 years (Figs. 2A–2C), stable during 40–69 years (Figs. 2D–2F), and then increasing again during 70–89 years (Figs. 2G and 2H). Conversely, as the blue arrows indicate, the frequencies of "some" response for 16 items were slightly increasing during

12–39 years (Figs. 2A–2C), decreasing during 40–69 years (Figs. 2D–2F) and were unclear during 70–89 years (Figs. 2G and 2H). It is necessary to quantify the age-related changes in the distributions of 16 negative items in order to evaluate the variations precisely.

With a log-normal scale, the distribution of the 16 negative symptom items for "some" to "most" responses showed a parallel linear pattern across all age groups, suggesting that they followed an exponential pattern with the same parameter $r$ (Fig. 3). Upon examining by age groups, the slopes of the lines for 16 negative items with a log-normal scale were seen to change according to age (12–19, 20–29, 30–39, 40–49, 50–59, 60–69, 70–79, and 80–89 years ) (Figs. 3A–3H). The slopes were more horizontal during 70–89 years (3G and 3H) than during 30–69 years (Figs. 3C–3F).

### Quantification of age-related changes in the distributions of 16 negative items

The parameters $r$ and $P$ were estimated from empirical data to quantify the age-related changes in the distributions of 16 negative items. Figure 4 shows that the average of the estimated parameter $r$ exhibited a U-shaped pattern, being high during 12–29 years, staying low during 30–59 years, and then increasing again during 60–89 years, similar to the U-shaped pattern of total CES-D score (Fig. 1). In contrast, the mean of the estimated parameter $P$ was stable across all age groups as compared with the parameter $r$. The average of estimated parameter $P$ slightly increased during 12–49 years, slightly decreased during 50–79 years, and then increased again during 80–89 years.

### Comparison of the Likert scoring and binary scoring

Finally, to demonstrate the effects of scoring methods in adulthood, total CES-D score and 16 item CESD scores were compared between the standard Likert (0,1,2,3) and binary methods (0-1-1-1) (Fig. 5). The total CES-D score and16 item CESD scores in the binary methods were estimated from empirical data. We analyzed the respondents between12–79 years as the older participants recruited by other studies were mainly under 80 years. In the standard Likert method, the total CES-D score and total 16–item score showed a U-shaped pattern (Fig. 5A). In contrast, in the binary method, the estimated total CES-D score and total 16-item score showed a downward trajectory with age (Fig. 5B). The total CES-D score and total 16-item score were higher in the 70–79 year group than the 30–59 year group in the Likert method, but lower in the 70–79 year group than the 30–59 year group in the binary method.

## DISCUSSION

The aim of the present study was to delineate the age-related changes in the distributions of each item score, and to test whether the trajectory of depressive symptoms varies with the scoring method.

The main findings of this study are that (1) the U-shaped pattern of total CES-D score across adulthood is mainly attributed to the age-related changes of 16 negative symptoms, (2) the mathematical model fits the distributions of 16 negative items across the adult span, and (3) the estimated parameter $r$ of 16 negative items showed a U-shaped pattern, being

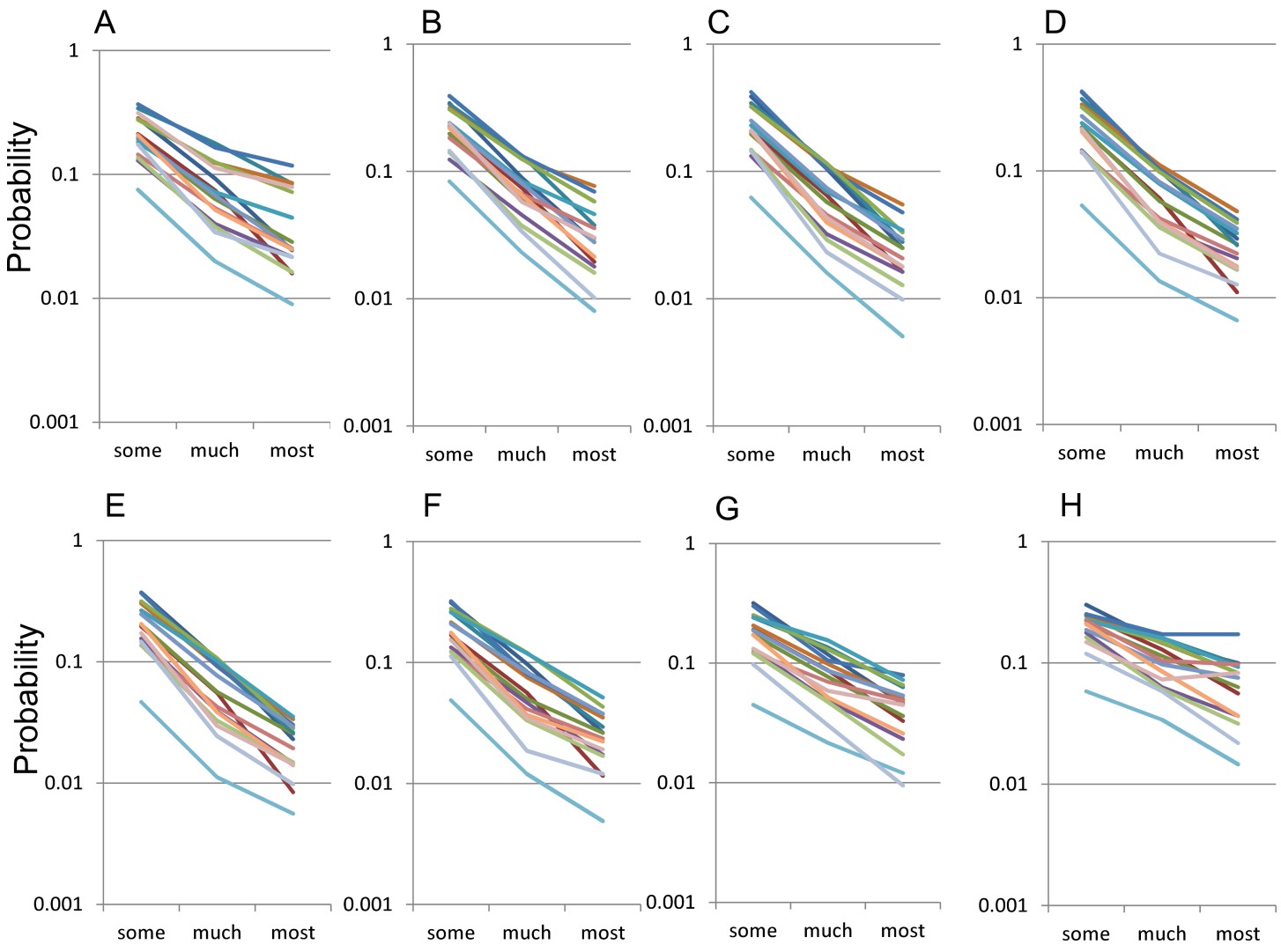

**Figure 3  The distributions of the 16 items for "some" to "most" responses by age group using a log-normal scale.** The distributions of the 16 items for "some" to "most" responses by age group, (A) 12–19, (B) 20–29, (C) 30–39, (D) 40–49, (E) 50–59, (F) 60–69, G) 70–79, and (H) 80–89 years. The items in this scale use the same color scheme as in Fig. 2. While the distribution of the 16 negative symptom items for "some" to "most" responses showed a parallel linear pattern across all age groups, the slopes of the lines for 16 negative items with a log-normal scale were seen to change according to age.

high during 12–29 years, staying low during 30–59 years, and then increasing again during 60–89 years.

The trajectory of depressive symptoms across adulthood could be explained by the aforementioned findings. Theoretically, the expected value of each negative item score is $P \times (3r^2 + 2r + 1)$ with the Likert method (0-1-2-3) and $P \times (r^2 + r + 1)$ with the binary method (0-1-1-1). Because expected values include the square of $r$ and the changes of parameter $r$ across the adult lifespan are more intense than that of parameter $P$, the trajectory of depressive symptoms are mostly affected by parameter $r$, resulting in the similar U-shaped patterns.

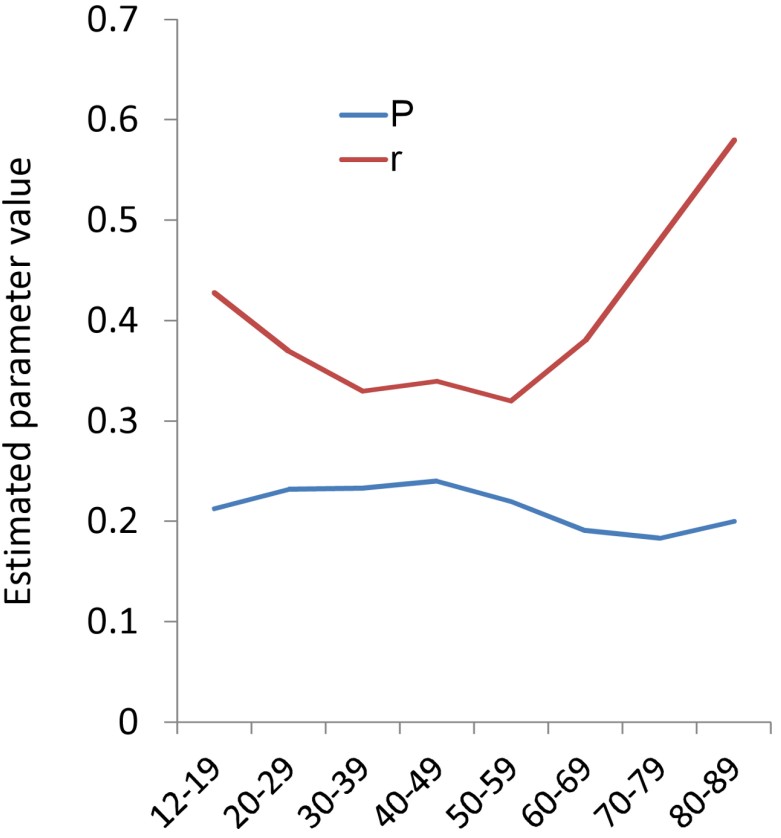

**Figure 4** **The relationships between age and estimated parameters *P* and *r*.** Red graph indicate the the average of the estimated parameter *r*. Blue graph indicate the probability of "some." The average of the estimated parameter *r* exhibited a U-shaped pattern, being high during 12–29 years, staying low during 30–59 years, and then increasing again during 60–89 years. In contrast, the average of parameter *P* slightly increased during 12–39 years, decreased during 40–79 years, and then slightly increased again during 80–89 years.

Our findings indicate that the increase of depressive symptoms for older age is based on the increase in parameter *r*, which corresponds to the increase of the probabilities of "much" and "most," suggesting that aging tends to induce more severe depressive symptoms. However, this explanation conflicts with the fact that most epidemiological evidence suggests that major depressive disorders decline with advancing age (*Kessler et al., 2010*). Generally, there is the discrepancy between the fact that rates of clinical depression are highest in midlife, whereas depressive symptom screening scale scores (using Likert method) are highest in older age (*Kessler et al., 1992*). It remains unclear whether people are more depressed with aging. One possible explanation is that there may be an age-related change in self-rating for the severity of depressive symptoms. It has been found that older individuals tend to have problems with retrieving information that is highly specific because they are less effective at controlling their retrieval process (*Luo & Craik, 2009*). Further studies are necessary to clarify the fact that depressive symptom screening scale scores are highest in older age.

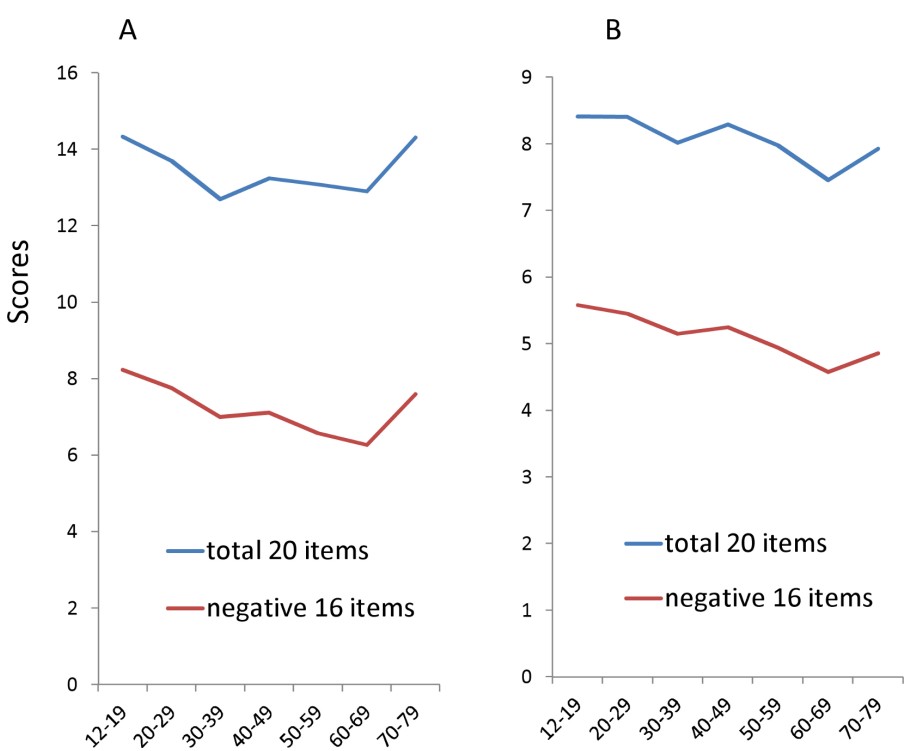

**Figure 5 The relationships between age and the total scores of 20 items and 16 negative items using the Likert and binary methods.** (A) In the standard Likert method, the total CES-D score and total 16–item score showed a U-shaped pattern. (B) In the binary method, the total CES-D score and total 16-item score showed a downward trajectory with age.

Although the findings of the present paper cannot explain the discrepancy between the rates of clinical depression and depressive symptom screening scale scores, they could explain the discrepancy between the Likert method and the binary method. Using the binary method, the total CES-D and 16-item scores were lower in the 70–79 age group than in the 30–59 age group (Fig. 5B). This finding could be explained as follows. First, although the parameter $r$ increases during 60–89 years, the expected value using the binary method does not reflect the increase in parameter $r$ as much as the Likert method. Secondly, the parameter $P$ decreases during 50–79 years, being lowest during 70–79 years. Thus, the CES-D score using binary method exhibits lower scores during 70–79 years than during 30–59 years (Fig. 5). Therefore, our studies support the hypothesis that the trajectory of depressive symptoms across the adult lifespan varies with the scoring method.

In addition to the scoring method, there is another methodological factor which may be responsible for the inconsistent evidence of the trajectory of depressive symptoms. While the total depressive symptom score follows a U-shaped pattern across adulthood, the age-related changes in depressive symptoms seem relatively mild (*Sutin et al., 2013*). The estimated average change per decade in CES-D total score is less than one point between the ages of 20 and 79 years, while the standard deviation of the CES-D scores in community surveys is between 5 and 10 points (*Kessler et al., 1992*; *Hek et al., 2013*; *Oh et al., 2013*). The mean total CES-D scores appear to stabilize between the ages of 30 and 69 years

 

in particular. As *Kessler et al. (1992)* pointed out, a number of studies have worked with samples having a truncated age range, a small number of very old respondents (older than 75 years), or a measure of age that combines all respondents over the age of 65 into a single group. These truncated age ranges may cause the researcher to overlook the non-linear increase in depressive symptoms that occurs after the age of 70 (*Newmann, 1989*; *Kessler et al., 1992*).

While 13 of the 16 negative symptom items exhibit U-shaped patterns across the adult lifespan, the remaining three negative items belonging to the somatic and retarded activities symptoms subscale do not exhibit this pattern. "Bothered," "sleep," and "talked" are the lowest during 12–19 years instead of 30–69 years, and the highest during 80–89 years. These results are congruent with previous reports that some items of the somatic and retarded activities symptoms subscale exhibit the lowest scores during young adulthood and the highest scores during older age (*Berry, Storandt & Coyne, 1984*; *Hegeman et al., 2012*).

In comparison with the 16 negative symptom items, the age-related changes in four positive affect symptom items were mild and differed from each other in the present study. Most of the studies report that the trajectories of positive affect symptom items differ from those of negative affect symptoms, although the evidence for trajectories of positive affects symptoms has been somewhat mixed (*Kunzmann, 2008*). While the age-related changes in positive affect items were mild, the total scores of 4 positive item scores increased during 30–89 years. This finding is in line with Carstensen's report that emotional well-being mildly improves from early adulthood to old age (*Carstensen et al., 2000*).

Recently, because of their relative independence, positive affect and negative affect have been commonly recognized as two different phenomena that should be studied individually (*Lucas, Diener & Suh, 1996*). A number of cross-cultural comparison studies have reported that the response patterns for the positive affects items vary according to ethnicity or nation (e.g., skewed, plateau-shaped, U-shaped and reverse U-shaped), whereas the response patterns for the 16 negative items were generally comparable (*Iwata, Roberts & Kawakami, 1995*; *Iwata et al., 1998*). Although the CES-D score is the composite score of the 20-item scores, it could be appropriate to recognize the 16 negative item scores and the positive scores as different scores (*Tomitaka, Kawasaki & Furukawa, 2015b*).

The present paper indicates that the mathematical model fits the distributions of 16 negative items across the adult span. The conditions that enable such a common distribution can be speculated upon. In general, self-rating depression scales are used as follows. First, each subject must determine whether a symptom is present. If the level of the symptom does not meet the threshold which excludes everyday mild symptoms, it is regarded as "rarely (less than 1 day)." Next, if a depressive symptom that meets the threshold is present, the duration of the symptom is quantified and divided into "some (1–2 days)," "much (3–4 days)," and "most (5–7 days)." This two-step process increases the possibility that "rarely" will cover the range that does not meet the threshold, while each of "some", "much", and "most" will cover the almost fixed ratio of the range that satisfies the threshold. If what we call the latent trait of depressive symptoms could be exponential distribution which has the memoryless property, the 16 negative symptom items for CES-D would commonly exhibit the mathematical distribution that was observed in the present study. In general,

exponential distribution is observed where individual variability and total stability are organized together, such as the Boltzmann–Gibbs law and income distribution (*Dragulescu & Yakovenko, 2000*). With respect to individual variability and total stability, the conditions that enable exponential distribution could be present in the distributions of the 16 negative items. Further consideration regarding this speculation is needed.

There are some methodological advantages in the present investigation. First, the sample was a representative of the general Japanese population, which reduced selection bias. The large sample size ($N = 21,040$) enabled us to elucidate the patterns of the distributions of depressive symptom items.

Second, the mathematical model was used to help understand the age-related changes in the distribution of depressive symptoms. Because these distributions are completely different from normal distribution, the parameters of normal distribution (e.g., average and standard deviation) were not sufficient to express the distributions themselves. After assessing the fit of the mathematical model to the distributions of depressive symptoms, we utilized it to quantify the age-related changes in the distributions of depressive symptoms. To the best of our knowledge, mathematical modeling is still not well developed in the field of psychiatry. While depressive symptoms of individuals are difficult to predict, the entire population may follow a certain mathematical pattern (*Tomitaka, Kawasaki & Furukawa, 2015a*).

This study has some limitations. First, a standard psychiatric diagnosis with structured interview was not performed for the subjects in this study. Therefore, the study did not encompass a psychiatric diagnosis of depressive symptoms. Second, although we evaluated the fit of the mathematical model to the distributions of depressive symptoms, analysis based on other mathematical models was not performed in this study. In general, the most important part of model evaluation is testing whether the model fits empirical data better than other models. However, to the best of our knowledge, no other mathematical models for the distributions of depressive symptoms have been reported so far. Despite these limitations, the present study provides important information regarding the age-related changes in depressive symptoms.

## CONCLUSION

Our results indicate that the scoring method of depressive symptoms is important in evaluating the age-related changes in depressive symptoms. The proposed mathematical model fits the distributions of depressive symptoms across adulthood, raising the possibility that depressive symptoms in the general population follow a mathematical pattern as a whole.

## ACKNOWLEDGEMENTS

The authors would like to thank the Active Survey of Health and Welfare project for providing the data for this study, Dr. Shinji Sakamoto for helpful advice, and Enago (www.enago.jp) for the English language review.

### Funding

The authors received no funding for this work.

### Competing Interests

The authors declare there are no competing interests.

### Author Contributions

- Shinichiro Tomitaka conceived and designed the experiments, performed the experiments, analyzed the data, wrote the paper, prepared figures and/or tables, reviewed drafts of the paper.
- Yohei Kawasaki, Kazuki Ide, Hiroshi Yamada, Toshiaki A. Furukawa and Yutaka Ono reviewed drafts of the paper.

### Human Ethics

The following information was supplied relating to ethical approvals (i.e., approving body and any reference numbers):

The present study was approved in 2014 by the ethics committee of Panasonic Health Center (approval number, 2014-1). The authors assert that all procedures contributing to this work comply with the ethical standards of the relevant national and institutional committees on human experimentation and with the Helsinki Declaration of 1975, as revised in 2008.

### Data Availability

The present study used data from the Active Survey of Health and Welfare (ASHW) conducted by the Japanese Ministry of Health, Labor and Welfare in 2000; however, the Japanese Ministry of Health, Labor and Welfare does not allow the publication of the data.

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
