# Peer review of "Age-related changes in the distributions of depressive symptom items in the general population: a cross-sectional study using the exponential distribution model"

_PeerJ, doi:10.7717/peerj.1547_

## Round 0.1 · original submission · Major Revisions

The two reviews are both thoughtful and make substantial practical recommendations. You can see that the issues are substantial, including statistical issues about factor structure, adequacy of the literature review, adequacy of the challenge made about using sums for scale values, and an apparent disinterest in the substantive issue of aging. Any one of these could lead to rejection, and a revision might not be acceptable. I am willing to consider a revision but cannot predict whether the paper will ultimately be acceptable because it is unclear to me whether you are able or willing to carry out what is essentially a reanalysis and rewriting.

Reviewer 1 ·

Basic reporting

This is an interesting study on the age specific changes of depression symptomatology using the CES-D as a measure of depression

Experimental design

A major criticism of mine to this manuscript is the authors’ idiosyncratic use of statistics. Using their past investigation, the authors presented a mathematical formula to predict probabilities of each of the CES-D items. Then they divided the CES-D items into the negative and positive items. Nevertheless, when using a measure with multiple items (such as the CES-D), researchers first shed light on its factor structure. In a meta-analysis of the factor analytic studies of the CES-D, Shafer et al. (2006) noted that the measure had a 4-factor structure: somatic, depressed affect, positive affect, and interpersonal problems. The authors of the present study should be encouraged to perform an exploratory factor analysis in a randomly divided sample and then a confirmatory factor analysis in the other halved sample. If the authors are not very happy about dealing the CES-D items as distributing normally, they should be referred to Muthen and Muthen’s Mplus method. This will give a clearer picture of the symptomatic structure of depression, If, as suggested by Shafer et al. (2006), the 4-factor structure is confirmed in the present study, the authors should discuss the age-specific effect on each of the four CES-D subscales. Another important topic is whether the factor structure of the CES-D is unvarying over the wide age ranges. The factor structure found among young adults may be different from that found among elderly people. This should be examined by a multi-group structural equation modelling method. These basic statistical examination should usher the main part of the study: age-specificity.

Shafer, A. B. (2006). Meta-analysis of the factor structures of four depression questionnaires: Beck, CES-D, Hamilton, and Zung. Journal of Clinical Psychology, 62, 123-146.

The problem of using both negative as well as positive items in order to measure depressive symptoms was clearly studied by Iwata and colleagues (1995, 1998). These reference should be carefully examined.

Iwata, N., Roberts, C. R., & Kawakami, N. (1995). Japan-U.S. comparison of responses to depression scale items among adult workers. Psychiatry Research, 58, 237-245.
Iwata, N., Saito, K., & Roberts, R. E. (1994). Responses to a self-administered depression scale among younger adolescents in Japan. Psychiatry Research, 53, 275-287.
Iwata, N., Umesue, M., Egashira, K., Hiro, H., Mizoue, T., Mishima, N., & Nagata, S. (1998). Can positive affect items be used to assess depressive disorders in the Japanese population? Psychological Medicine, 28, 153-158.


Others comments are as follows:

1: Was the ethical approval given prior to 2000 when the Activity Survey of Health and Welfare (ASHW) was conducted? Why did the Panasonic Health Centre, seemingly a private sector, give ethical approval to this kind of a government-led survey?

2: More than 9,000 participants’ data were deleted list-wise when any one CES-D items, age, or sex were missing. There is a great statistical concern about producing a bias in the study’s results. He authors should perform multiple imputation for dealing with cases with variables with missing data.

3: The abstract should be shortened.

4: Comparison of the Likert and binary methods lack scientific ground. Why should this be examined? I do not see any merit in such a comparison.

Validity of the findings

This manuscript has strength in its sample size. However, it requires further consideration in statistics.

·

Basic reporting

Review: Age-related changes in depressive symptom items across the life-span in the general population…
Strengths: This paper undertakes a puzzling inconsistency in the measurement of depressive symptoms in the elderly and tries to resolve it with mathematical analysis of the probability of response choice of older people on a Likert-like scale. The writing is excellent; the figures enhance the text, the sample has the power to address the research questions.

Experimental design

As far as I could make out the experimental design was well-suited to address the research questions.

Validity of the findings

Weaknesses:
As a general criticism, this paper is written as if the authors have no stake in the actual issue of ageing and depression. This makes the paper much less interesting than it can be. Here are some specific points that may be addressed by the authors:
1. The literature review does not make it clear if this a novel approach or whether others have undertaken it.
2. The idea that a Likert-like scale is not an interval scale but an ordinal scale is at the base of the approach of the current work. It should be spelled out and discussed.
3. If the authors are correct, then the practice of calculating a scale sum or scale mean and using that number as a continuous measure of the construct in question is wrong. Since this is widely used in psychological research and in social science in general the discussion really needs to face this head-on, and discuss the implications of the results.
4. I was left unclear about what the authors conclude: do they conclude that with ageing people are more depressed? Or are they saying that this is a response bias that comes with ageing and therefore an artifact? Or are they saying that it is impossible to tell which of the two is correct?
5. Another point that comes up in the Results but is not really addressed in the Discussion, is the fact that as depressive symptoms are more endorsed by the elderly so are the positive affect items. Is this an artifact? Is this part of the ageing happiness paradox that Laura Carsten has studied?

Additional comments

As a general criticism, this paper is written as if the authors have no stake in the actual issue of ageing and depression. This makes the paper much less interesting than it can be.

---

## Round 0.2 · Minor Revisions

Your response to the earlier reviews was excellent and clarified the purpose and significance of your findings. I would encourage you to consider shortening the abstract to 250 words since some abstracting services truncate abstracts after 250 words. Your current abstract is over 400 words. Also notice that "aging" is misspelled in your revision - there is no "e" before "i". Also referring to a "mathematical model" is not informative; I suggest it would be better to refer to an exponential distribution explicitly. Thank you for your useful observations and conclusions.

---

## Round 0.3 · accepted · Accept

Your thoughtful article is a valuable contribution to the literature.